DagsHab
x ML@Purdue
Hackathon

Study

# Ablation Studies on "TextPSG: Panoptic Scene Graph Generation from Textual Descriptions - ICCV 2023"

*** ***[1], *** ***[2]*** ***[3]

[1]*** – [2]*** – [3]***

Written by
*** ***
*** ***
*** ***

Received
04 November 2023

### TextPSG Reproducibility Study

Semantic representation and grouping of objects are extremely critical in deciphering image scenes. While traditional end-to-end models often employ a top down approach, extracting and segmenting images from pixel annotations, this approach is costly and tedious, leading to limited datasets that are hard to obtain. In contrast, more recent models such as TextPSG aim to eliminate this problem by leveraging large, pre-existing datasets of image-caption pairs in order to generate Panoptic Scene Graphs (PSGs), collecting no pre-existing location priors, explicit links between visual and textual entities, or concept sets. In this work, we aim to reproduce TextPSG's claims in order to determine (1) the ease of reproducibility and (2) perform ablation studies to discover the most impactful parameters of the model[1].

## 1 Reproducibility

There are four principal components to TextPSG: The region grouper, entity grounder, segment merger and label generator. The region grouper aims to group pixel regions in the image into various segments. Each segment ideally represents a singular entity. The image caption is pre-processed into a text graph, which extracts the nouns/entities in the caption. Each entity in the pre-processed caption is then aligned with the image embedding segments generated by the region grouper. To facilitate this alignment is the role of the entity grounder, which projects the image segments and entity embeddings into a shared feature space, then performing fine grained contrastive loss. The alignment results then help the segment merger determine the segment similarity to assign the final segmentations and the label generator extracts and predicts semantic information. As no existing code base or pseudo-code is provided for this paper, we used our best judgement for details not explicitly given by the paper.

### 1.1 The Region Grouper

Following the TextPSG implementation, we employ a pre-trained GroupViT model. GroupViT first samples $N$ non-overlapping patches [2]. The model then undergoes 2 stages of grouping. In the first stage, the $N$ patches are grouped into 64 unique segments. In the second stage, those 64 unique segments are further grouped into 8 unique segments. In our test, we extract the 64 image segment embeddings from the first stage as an input to our entity grounder.

## 1.2 Text Embeddings

While image embeddings can be extracted directly from GroupViT, text embeddings require preprocessing to extract the individual entities in the caption. Following the TextPSG implementation, we employ two methods to generate a directed text graph, with entities as nodes and relations as edges. We make use of OpenIE and the StanfordNLP library to generate two separate text graphs and then take their union to generate one text graph. In total, $k$ entities are generated from the text preprocessing. Fig 1. provides a visualization of a text graph. After the preprocessing, an embedding vector for each caption is generated via the pre-trained text transformer in GroupViT, $\mathbf{Tfm}^T$. Coupled with the entities from the text graph, $k$ sentences are generated by appending the entity to the end of the caption. The $k$ sentences are embedded by $\mathbf{Tfm}^T$, creating an output tensor of dimension $k \times s \times 196$, where $s$ is the number of tokens in the embedding. Finally, these vectors are padded to dimension $20 \times 20 \times 196$, generating the final text embeddings.

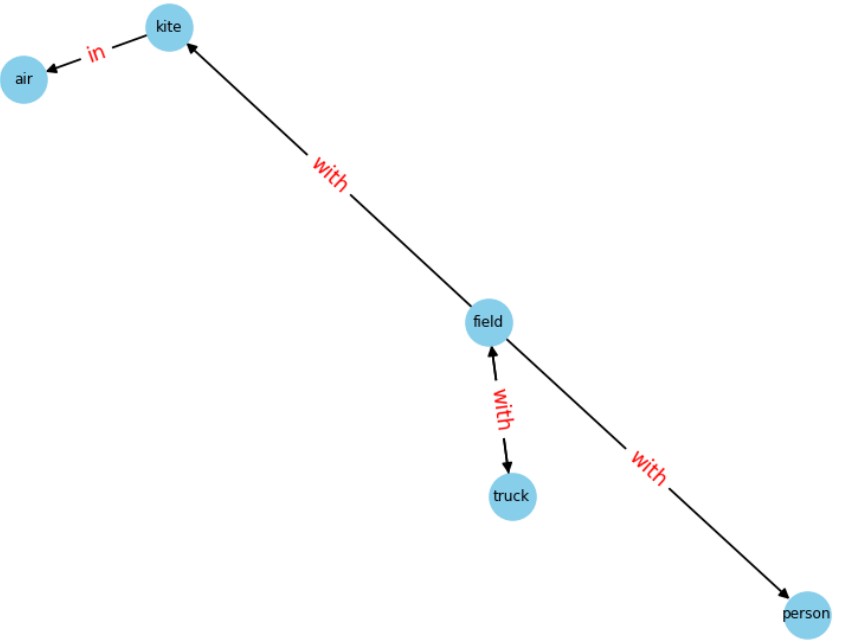

**Figure 1:** Text graph for caption: A truck driving through a field with people flying kites.

## 1.3 The Entity Grounder

Following the TextPSG implementation, first an MLP is used to project the segment and entity embeddings into a shared feature space $\mathcal{F}$. As details on the MLP are not provided in the paper, we used the GELU activation function. For the contrastive loss, we first calculate the fine grained loss from the segment embeddings in the group $x^k$ to all the text embeddings $y$ across batches $B$. This functions by taking the average of the filtered maximum cosine-similarity values (filtered by applying a filtering threshold $\theta$) of the text embeddings in one sample to every text embedding across samples in the batch. This follows the equations

$$p_i^k = \max_{1 \le j \le E} \cos[x_i^k, y_j]$$

and

$$p^k = \frac{1}{\sum_{i=1}^{H_k} 1_{p_i^k > \theta}} \sum_{i=1}^{H_k} (p_i^k \cdot 1_{p_i^k > \theta})$$

In the paper, the fine-grained contrastive loss from Image to Text is calculated as:

$$\mathcal{L}_{fine}^{k,I \to T} = -\frac{1}{B} \sum_{i=1}^{B} \frac{\exp\left(\frac{p^{k,i \to i}}{\tau}\right)}{\sum_{j=1}^{B} \exp\left(\frac{p^{k,i \to j}}{\tau}\right)}$$

Conceptually, this is the negative mean of the exponentiation of the average maximum cosine similarities of the segment embeddings in one sample compared the text embeddings in all the samples, and decreasing this loss function entails maximizing the similarity within one sample while minimizing similarity with unrelated samples, thus serving as an alignment stage. The temperature tau is just explained as a learnable temperature. However, we found that this loss was not functioning as intended, returning a near constant value and we found the problem to be the lack of the logarithm after the division of the exponents which is presented in the fine grained contrastive loss implementation in FILIP[3], and so we added the logarithm in the summation giving us a loss function:

$$\mathcal{L}_{fine}^{k,I \to T} = -\frac{1}{B} \sum_{i=1}^{B} \log\left(\frac{\exp\left(\frac{p^{k,i \to i}}{\tau}\right)}{\sum_{j=1}^{B} \exp\left(\frac{p^{k,i \to j}}{\tau}\right)}\right)$$

The fine-grained contrastive loss from entity embeddings to segment embeddings $\mathcal{L}_{fine}^{T \to I}$ follows a similar set of operations

$$q_i^k = \max_{1 \le i \le H_k} \cos[x_i^k, y_j]$$

$$q^k = \frac{1}{\sum_{j=1}^{E} 1_{q_j^k > \theta}} \sum_{j=1}^{E} (q_j^k \cdot 1_{q_j^k > \theta})$$

$$\mathcal{L}_{fine}^{k,T \to I} = -\frac{1}{B} \sum_{i=1}^{B} \log\left(\frac{\exp\left(\frac{q^{k,i \to i}}{\tau}\right)}{\sum_{j=1}^{B} \exp\left(\frac{q^{k,i \to j}}{\tau}\right)}\right)$$

Thus, the total fine-grained contrastive loss is the average of these two losses:

$$\mathcal{L}_{fine} = \frac{1}{2}(\mathcal{L}_{fine}^{k,I \to T} + \mathcal{L}_{fine}^{k,T \to I})$$

**Segment Merger –** Following the implementation of the segment merger in TextPSG, we use the entity grounder to supervise the learning of a group of similarity matrices which are used for a smaller stage segment merging. For every segment in group $k$, we compute the pairwise cosine similarity between image segments which is then re-scaled to $[0, 1]$ giving us a similarity matrix $\mathbf{Sim}_k \in [0, 1]^{H_k \times H_k}$. The formula for the similarity matrix ends up looking like $\mathbf{Sim}_k[i, j] = \frac{1}{2}(\cos[x_i^k, x_j^k])$. Subsequently, as directed by the paper, we use the pseudo labels to create a target similarity matrix where for every label for every segment, we set the target similarity to 1 if the pseudo labels $l_i^k$ equals $l_j^k$ and if the similarity between segments and their pseudo labels is above the filtering threshold, otherwise we set the value to 0. This looks like:

$$\mathbf{Sim}_k^{target}[i, j] = \begin{cases} 1, \text{if } l_i^k = l_j^k \ \ \& \ \ \cos[x_i^k, y_{l_i^k}] > \theta \ \ \& \ \ \cos[x_j^k, y_{l_j^k}].\theta, \\ 0, \text{otherwise} \end{cases}$$

As TextPSG formulates, the similarity loss ends up looking like:

$$\mathcal{L}_{sim}^k = \frac{1}{H_k^2} ||\mathbf{Sim}_k - \mathbf{Sim}_k^{target}||_F^2$$

## 2 Results

### 2.1 Qualitative Results

We examine the qualitative results of the Entity Grounder below in Figure 2. Darker patches are the image segments that the models believe are more closely aligned with the noun/entity segments. Overall, our model managed to correctly identify some of the image segments. We also observed that the model would sometimes pair the entity embedding with the inverse of the object. The last column of the figure demonstrates the cosine similarity between each image segment and the entity segment. A higher cosine similarity indicates a stronger match.

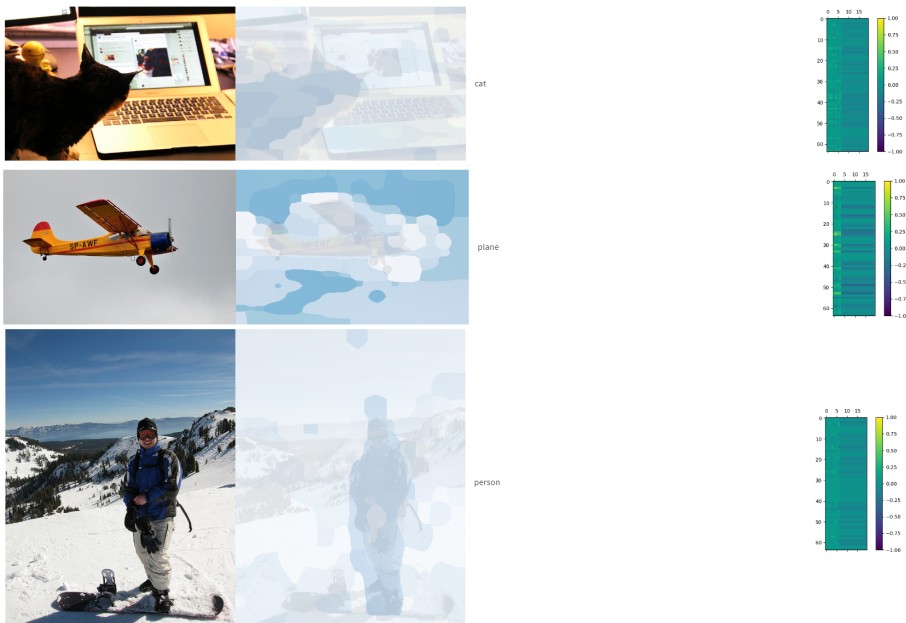

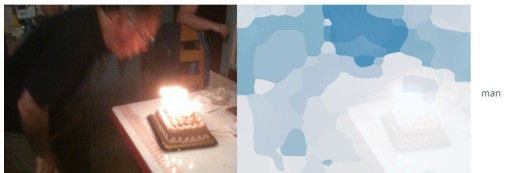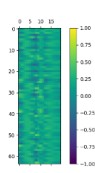

**Figure 2:** Examples of entity grounder outputs generated by our model. The heat map column highlights the image segments the model believes corresponds to the entity/noun in the third column. The final column outputs the cosine similarity between each image segment and the entity segment.

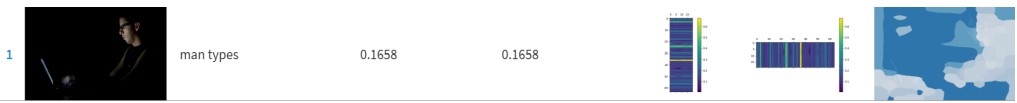

**Figure 3:** Examples from alternate model architecture and run-time showing inverted cosine similarity masks.

## 2.2 Ablation Studies

**Activation Function –** We experimented with five activation functions: ReLU, GELU, SiLU, Tanh, and Sigmoid. We found the ReLU resulted in catastrophic failure as it zeroed out our input tensors due to a prevalence of negative numbers. We also saw the Tanh and Sigmoid performed about the same, neither being particularly useful because it projected the tensors into too small a space and resulted in a loss of fidelity. We found that GELU, or SiLU, worked the best as it was a simple nonlinearity that did not project negative values to zero and remained nearly linear for positive values.

**RNN and MLP –** We tested different frameworks for both the RNN and MLP. For the RNN, we tested dimensionality reduction by (1) taking the weighted sum of each segment embedding, and (2) simply flattening each segment-embedding pair. Both models perform comparably; however, the weighted sum was ultimately implemented to match with the TextPSG implementation. We also implemented MLP models with 1, 4, and 6 hidden layers. The model with 1 hidden layer had the best performance. We hypothesize that the models with 4 and 6 hidden layers suffered from being too complex.

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
