# OpenReview forum: "Ablation Studies on ”TextPSG: Panoptic Scene Graph Generation from Textual Descriptions - ICCV 2023”"
_purdue.edu/Purdue_University/ML/2023/Hackathon_Reproducibility_Challenge — Purdue University ML 2023 Hackathon Reproducibility Challenge Submission_

### Official Review · Reviewer_JFSz · 2023-12-01
**Review of Ablation Studies on ”TextPSG: Panoptic Scene Graph Generation from Textual Descriptions”**

**Rating:** 6
**Confidence:** 4

**Review:**

Page
1. Reproducibility explained
    1. I like the attempt a lot, but there are lots of visuals that are missing in this whole thing. How would we be optimizing the text graph generated, and how do losses look within training? The goal is reproducibility but we have no insight into how this would operate
    2. No localization prior? The textPSG specifically mentioned how they used a detector PRIOR to running vision transformers. There is no mention of what this additional detector would be (clarify if its the groupViT here).
    3. Segment merger learned similarities between segments to merge stuff while the label generator learns relationship b/w object semantics and the verb predicates. It is even mentioned in the reproducibility intro as a core part of the project, yet there is no mention of a label segmented
2. Results
    1. From a quick glance textPSG seems that the core of their approach is that we can have semantic representation comes from just images and text pairing with it. The direction of showing heat maps and cosine similarities for the first three components is very helpful, but not being able to see all three work in tandem makes it hard to see how mnuch was reproduced
    2. Out to distribution dataset? textPSG takes pride on explaining how their performance compares on OOD datasets, yet there is no mention of what dataset they trained on at all.